# Phenolic and Metabolic Profiles, Antioxidant Activities, Glycemic Control, and Anti-Inflammatory Activity of Three Thai Papaya Cultivar Leaves

**DOI:** 10.3390/foods13111692

**Published:** 2024-05-28

**Authors:** Sirinet Chaijan, Manat Chaijan, Umaporn Uawisetwathana, Atikorn Panya, Natthaporn Phonsatta, Kalidas Shetty, Worawan Panpipat

**Affiliations:** 1Food Technology and Innovation Research Center of Excellence, Department of Food Science and Innovation, School of Agricultural Technology and Food Industry, Walailak University, Nakhon Si Thammarat 80160, Thailand; stsirinet.ch@gmail.com (S.C.); cmanat@wu.ac.th (M.C.); 2Microarray Research Team, National Center for Genetic Engineering and Biotechnology (BIOTEC), National Science and Technology Development Agency (NSTDA), Khlong Nueng, Khlong Luang, Pathum Thani 12120, Thailand; umaporn.uaw@biotec.or.th; 3International Joint Research Center on Food Security (IJC-FOODSEC), 111 Thailand Science Park, Pahonyothin Road, Khong Luang, Pathum Thani 12120, Thailand; 4Thailand Metabolomics Society, Bangkok, Thailand; 5Food Biotechnology Research Unit, National Center for Genetic Engineering and Biotechnology (BIOTEC), 113 Thailand Science Park, Phaholyothin Rd., Khlong Nueng, Khlong Luang, Pathum Thani 12120, Thailand; atikorn.pan@biotec.or.th (A.P.); natthaporn.pho@biotec.or.th (N.P.); 6Global Institute of Food Security and International Agriculture (GIFSIA), North Dakota State University, 374 D Loftsgard Hall, 1360 Albrecht Blvd., Fargo, ND 58108, USA; kalidas.shetty@ndsu.edu

**Keywords:** *Carica* *papaya*, papaya leaf extract, metabolic profile, in vitro biological activity

## Abstract

This study thoroughly examined the proximate composition, bioactive composition, and in vitro biological activities of three different cultivars of papaya leaf extracts (PLEs) as potential functional ingredients and nutraceuticals. The dark green leaves of three papaya cultivars, Khaek Dam (KD), Holland (H), and Thai Local (L), were used in this study. The protein content of the leaves ranged from 25.96 to 32.18%, the fat content ranged from 7.34 to 11.66%, the carbohydrate content ranged from 5.80 to 17.91%, the moisture content ranged from 6.02 to 6.49%, the ash content ranged from 11.23 to 12.40%, and the fiber content ranged from 23.24 to 38.48%. The L cultivar possessed significantly higher protein and carbohydrate contents, whereas the H cultivar had the highest ash content (*p* < 0.05). The total phenolic content (TPC) ranged from 113.94 to 173.69 mg GAE/g extract, with the KD cultivar having the highest TPC (*p* < 0.05). Several metabolic compounds such as phenolic compounds (particularly kaempferol, isorhamnetin, quercetin, ferulic acid, isoferulic acid, salicylic acid, sinapic acid, syringic acid, and vanillin), terpenoids (such as eucalyptol), glycosides, and indole were identified. The PLE from the KD cultivar had the highest levels of DPPH^•^ inhibition, metal chelation, reducing power, and antidiabetic activity (*p* < 0.05), suggesting superior biological activity. All three PLEs reduced the proliferation of RAW 264.7 cells in a dose-dependent manner with low nitric oxide formation. These results indicate that the papaya leaf, particularly from the KD cultivar, could be a promising source of functional food ingredients.

## 1. Introduction

The challenge and opportunities of discovering effective plant extracts for disease prevention or to boost human health and developing them as sources of functional food ingredients has worldwide interest. *Carica papaya* L. (papaw or papaya) is an herbaceous fruit-bearing plant belonging to the family Caricaceae [1]. Papaya is grown well in both tropical and subtropical regions, with its origins in Central America and Mexico [1]. Papaya, also referred to as “Ma La Gor”, is a significant fruit crop in Thailand, with an estimated 87 metric tons of papaya fruits produced in 2019 [2]. In Thailand, there are several commercially available and popular papaya cultivars, including “Khaek Dum”, “Khaek Nuan”, “Coco”, “Sai Nam Phueng”, “Pak Chong”, “Hawaii”, and “Pak Mai Lai, or “Holland” [3,4]. 

Aside from its use as fruit, papaya is used in traditional medicine due to its secondary metabolite activities with therapeutic potential. Because papaya leaves exhibit antioxidant activity and also possess anti-inflammatory, anticancer, potential antiviral dengue prevention, antitumor, and antimalarial activities, as well as protection against oxidative damage, they could be developed as functional food ingredients and for nutraceutical applications as well as designed for therapeutic benefits [5,6,7,8]. Papaya leaves (PLs) contain a variety of phytochemical compounds and bioactive components, including papain, flavonoids, and polyphenols. They also contain alkaloids, tannins, terpenoids, saponins, chymopapain, α-tocopherol, cystatin, ascorbic acid, glucosinolates, cyanogenic glucosides, *p*-coumaric acid, and caffeic acid [9]. These phytochemicals have antioxidant, antimicrobial, and important pharmacological properties [6]. PLs are usually discarded, but there is promising potential to develop them as valuable sources of phytochemicals and bioactive components that may be employed in therapeutic and health beneficial formulations for the food and cosmetics industries.

Since PLs were used as a traditional herb to treat ailments, several studies have measured and identified the bioactive compound found in PLs. PLs typically contain a wide range of health-promoting phytochemicals, including alkaloids, saponins, glycosides, flavonoids, phenolic compounds, enzymes, amino acids, lipids, carbohydrates, vitamins, and minerals [10]. Several phenolic compounds have been identified in papaya leaf extract (PLE), such as quercetin, kaempferol 3-rutinoside, quercetin3-(2G-rhamnosylrutinoside), myricetin 3-rhamnoside, caffeic acid, *p*-coumaric acid, ferulic acid, protocatechuic acid, chlorogenic acid, and other derivatives [11,12].

In the study of Soib et al. [13], phytochemicals in PLE were identified such as carpaine, kaempferol 3-(2G-glucosylrutinoside), kaempferol 3-(2″-rhamnosylgalactoside) 7-rhamnoside, kaempferol 3-rhamnosyl-(1->2)-galactoside-7-rhamnoside, luteolin 7-galactosyl-(1->6)-galactoside, orientin 7-*O*-rhamnoside, 11-hydroxy-12,13-epoxy-9-octadecenoic acid, palmitic amide, and 2-hexaprenyl-6-methoxyphenol. It is believed that the anti-inflammatory and anticancer properties of PLs are primarily attributed to alkaloids, saponins, glycosides, phenolic compounds, and flavonoids [14]. The most important health-promoting and major bioactive components found in PL are carpaine, dehydrocarpaine I, and dehydrocarpaine II [15]. The aforementioned alkaloids have the ability to reduce high blood pressure and rapid heart rate and are effective for uterus relaxation, bronchiole dilatation, and movement of the intestinal strips, as well as antiplasmodial treatment [16]. Important phytochemical constituents of PLs were linked to their functional properties and structures. Furthermore, bioactive constituents, contents, and biological activities varied between papaya species. Nisa et al. [17] observed that Grendel and purple PL had higher total flavonoid content (50.33 and 46.02 µg/g, respectively) and had a higher percentage of radical scavenging properties than other cultivars such as Bangkok, California, and Golden. In the extract of PLs of the ‘Red Lady’ cultivar, quercetin and its derivatives showed higher antioxidant activity than kaempferol and its derivatives [18]. The bioactive contents, constituents, and biological activities of PL varied according to the papaya cultivar. It is of great interest to identify and analyze bioactive compounds associated with biological activities in PL from various papaya cultivars that could be useful in the use of wastes and byproducts from commercial PLs.

Based on the above rationale, and to advance the continuing scientific need to fill the gap in the limited use of PLs, there is an essential need to investigate the bioactive substances and associated biological activities of PLs. Thus, in this study, a thorough investigation into the extraction of bioactive components associated with biological activities was analyzed to increase the value of agricultural waste generated for PLs. The overall purpose of this study was to investigate how different papaya cultivars affected the phenolic composition, metabolic composition, antioxidant activities, antidiabetes properties, cytotoxicity, and anti-inflammatory activity of three PLEs harvested in southern Thailand.

## 2. Materials and Methods

### 2.1. Plant Materials

Papaya leaves (PLs) were harvested from a papaya farm in Nakhon Si Thammarat, Thailand, during the period from May to August 2023. The three commercial PL cultivars used were Khaek Dam (KD), Holland (H), and Thai Local (L). The PLs were washed, cut into small pieces, and dried in a tray dryer at 45 °C for 72 h (reduced to moisture content <10%, *w*/*w*). The dried PLs were crushed and passed through an 80-mesh sieve. The PL powder was vacuum-packed in polythene bags and kept at −20 °C until used.

### 2.2. Determination of Proximate Composition of PL Powder

The proximate composition of the PL powder was determined using the AOAC [19] standard methods. Moisture (AOAC method number 950.46), crude protein (AOAC method number 928.08), ash (AOAC method number 920.153), fiber (AOAC method number 962.09), fat (AOAC method number 963.15), and carbohydrate (calculated by difference) were all assessed.

### 2.3. Preparation of Papaya Leaf Extract (PLE)

The PLE was produced using the procedures described by Adenowo et al. [20] and Gomez-Estaca et al. [21], with some minor modifications. The dried PL powder was soaked in absolute ethanol at room temperature (27–29 °C) for 24 h, with a solid-to-solvent ratio of 1:10 (*w*/*v*). The solution was then filtered through Whatman No.1 filter paper. The filtrate, referred to as PLE, was concentrated using a vacuum rotary evaporator at 40 °C. All PLE were vacuum-packed and stored at −20 °C for further investigation.

### 2.4. Determination of Bioactive Constitutes of PLE

#### 2.4.1. Total Phenolic Content (TPC)

The TPC of PLE was evaluated using the Folin–Ciocalteu reagent method according to Irondi et al. [22], with some modifications. Briefly, 0.1 mL of PLE was mixed with 1.5 mL of Folin–Ciocalteu reagent (Sigma Aldrich, St. Louis, MO, USA).), which was previously diluted 10-fold with deionized water. Thereafter, 1.2 mL of 7.5% sodium carbonate solution (Sigma Aldrich) was added. The corresponding solution was incubated at room temperature in the dark for 30 min. The absorbance was measured at 765 nm and gallic acid was used as the standard. TPC was expressed as mg gallic acid equivalent (GAE) per gram of extract (mg GAE/g).

#### 2.4.2. Determination of Total Flavonoid Content (TFC)

The TFC was measured using the method described by Pothitirat et al. [23]. Briefly, 0.5 mL of PLE was combined with 1.5 mL of 95% ethanol and 0.1 mL of 10% aluminum chloride (Sigma Aldrich). Then, 0.1 mL of 1 M potassium acetate and 2.8 mL of distilled water were mixed. The absorbance at 415 nm was measured after 30 min of incubation at room temperature. TFC was expressed as milligrams of rutin equivalents (RE) per gram of extract.

#### 2.4.3. Determination of the Tannin Content (TC)

TC was analyzed with the method of Makkar et al. [24] using insoluble polyvinyl-polypirrolidone (PVPP). Briefly, 1 mL of PLE was mixed with 100 mg of PVPP. The mixture was centrifuged at 3000× *g* at 4 °C for 10 min after incubation at 4 °C for 15 min. The non-tannin phenolic supernatant was used for Folin–Ciocalteu analysis, following the TPC procedures described previously. TC was calculated as the difference between total and non-tannin phenolic content.

### 2.5. Analysis of Metabolic Profiles of PLE Using Liquid Chromatography-High-Resolution Tandem Mass Spectrometry (LC-HRMS/MS)

The metabolic profiles of PLEs were determined using a UPLC Dionex UltiMate 3000 RS system (Thermo Fisher Scientific, Waltham, MA, USA) connected to a high-resolution mass spectrometer (Orbitrap Fusion^TM^ Tribrid^TM^, Waltham, MA, USA), as described by Uawisetwathana et al. [25]. The chromatographic separation was performed on a Waters Acquity UPLC^®^ HSS T3 column (100 mm, 2.1 mm inner diameter, 1.8 μm particle size) at 35 °C. Water (solvent A) and methanol (solvent B) served as mobile phases for separation, and both were acidified with 0.1% formic acid. The elution gradient started with an isocratic step at 0–0.5 min (1% solvent B) at a flow rate of 0.300 μL/min along the separation, then progressed to 0.5–8.0 min (20% solvent B), 8.0–10.0 min (40% solvent B), 10.0–21.0 min (99% solvent B), and was held for 1 min. The system was then re-equilibrated to its initial state for 1.0 min (1% solvent B) and held for 2 min before starting the next cycle.

The mass spectral data were collected using electrospray positive and negative ionization modes with the following parameters: 3500 V capillary voltage, 333 °C ion transfer temperature, and 317 °C vaporizer temperature. Using a 120,000 resolution Orbitrap detector, a full scan data-dependent tandem mass spectrometry (full scan dd-MS^2^) approach was used to record MS and MS^2^ spectra spanning the mass range of 50–1200 Da. The ten most intense ions from the chromatographic runs were fragmented with higher-energy collisional dissociation (HCD) at 45% collision energy and 15,000 resolutions.

The acquired MS data from each sample were processed with Compound Discoverer (CD) 3.1.0 software (Thermo Scientific, Waltham, MA, USA) with the following parameters: (i) retention time alignment, (ii) unknown compound detection, (iii) elemental compositions prediction, and (iv) compound annotation against public databases and the mzCloud fragmentation database with mass tolerance at 5 and 10 ppm, respectively. Areas under picked peaks of the aligned chromatograms of all the examined samples were obtained and were normalized to their total abundance. The abundance of metabolite profiles among three different PL varieties were subjected to principal component analysis to see their differences. Then, the metabolite profiles were processed by selecting for (i) the % coefficient of variation (%CV) between replicates was less than 60% of the three varieties and (ii) the annotation confidence level of elemental composition and precursor mass was greater than 80%. Putative metabolites obtained from the annotation were classified based on their chemical formula, molecular mass and mass fragmentation. 

### 2.6. Determination of Antioxidant Activities of PLE

#### 2.6.1. DPPH Radical-Scavenging Activity

The DPPH^•^ scavenging activity of the PLEs was determined using the Fernández et al. [26] method, with minor modifications. In brief, 80 µL of PLE was mixed with 0.32 mL of 95% ethanol. Next, 2 mL of 0.06 mM DPPH methanol solution was added. After 30 min of incubation at room temperature in the dark, the absorbance at 517 nm was measured. A positive control, α-tocopherol, was used. The DPPH^•^ scavenging activity was calculated using the following equation, and the IC_50_ value was expressed.
(1)$${\mathrm{DPPH}}^{•}\;\mathrm{scavenging}\;\mathrm{activity}\;\left(\%\right)=\frac{\left({\mathrm{Abs}}_{\mathrm{control}}-\;{\mathrm{Abs}}_{\mathrm{sample}}\right)}{{\mathrm{Abs}}_{\mathrm{control}}}\times 100$$

#### 2.6.2. Reducing Power

The Prussian blue assay was used to determine reducing power, with slight modifications based on Canabady-Rochelle et al. [27]. PLEs were diluted tenfold with a 0.2 M phosphate buffer (pH 6.6). Then, 105 μL of 1% potassium ferricyanide was added to the extracts (210 μL). The mixture was incubated at 50 °C for 20 min. After adding 405 μL of distilled water, 10% trichloroacetic acid (99 μL), and 0.1% ferric chloride (81 μL), the mixture was incubated at room temperature for 10 min. The absorbance at 700 nm was measured. Ascorbic acid was used for reference. The reducing power was expressed as milligrams of ascorbic acid equivalents (AAE) per gram of extract.

#### 2.6.3. Chelating Activity

The chelating activity was determined using the methods described by Dinis et al. [28] and Turan et al. [29]. To summarize, 0.5 mL of 0.1 mM FeSO_4_ and 0.7 mL of 0.25 mM ferrozine were added to 1 mL of PLE. After a 10 min incubation at room temperature, the absorbance was measured at 562 nm. The standard was ethylene diamine tetra-acetic acid (EDTA), and an IC_50_ value was provided. The Fe^2+^ chelating activity was computed as follows:(2)$$\mathrm{Chelating}\;\mathrm{ability}\;\left(\%\right)=\frac{\left({\mathrm{Abs}}_{\mathrm{control}}-\;{\mathrm{Abs}}_{\mathrm{sample}}\right)}{{\mathrm{Abs}}_{\mathrm{control}}}\times 100$$

### 2.7. Determination of In Vitro Glycemic Index

#### 2.7.1. In Vitro α-Amylase Inhibitory Activity

The in vitro α-amylase inhibitory activity was conducted using a slightly modified method offered by Kim et al. [30] and Yu et al. [31]. Briefly, 0.2 mL of the PLE were mixed with 0.5 mL of 100 mM sodium phosphate buffer (pH 6.9) plus 6.7 mM sodium chloride and 0.2 mL of α-amylase solution (1 unit/mL, 0.1 M potassium phosphate buffer, pH 6.9) (Sigma Aldrich). After 20 min of incubation at 37 °C, the reaction was started by adding 0.2 mL of 1% starch solution in sodium phosphate buffer (pH 6.9). The mixture was allowed to stand at 37 °C for an additional 30 min. After adding 1 mL of DNS reagent and heating in a water bath at 100 °C for 10 min, the reaction was stopped by cooling to room temperature and adding 5 mL of distilled water. The absorbance was measured at 540 nanometers. Pure acarbose served as a positive control. To determine the IC_50_ values, inhibitory activity of PLE against α-amylase was tested at different concentrations in three replications. α-Amylase inhibition was calculated as follows:(3)$$\alpha -\mathrm{amylase}\;\mathrm{inhibitory}\;\mathrm{activity}\;\left(\%\right)=\frac{\left({\mathrm{Abs}}_{\mathrm{control}}-\;{\mathrm{Abs}}_{\mathrm{sample}}\right)}{{\mathrm{Abs}}_{\mathrm{control}}}\times 100$$

#### 2.7.2. In Vitro α-Glucosidase Inhibitory Activity

The in vitro α-glucosidase inhibitory activity was measured using the method described by Agada et al. [32], with slight modifications. The reaction mixture included 20 μL of various PLE concentrations and 50 μL of α-glucosidase solution (1 unit/mL, 10 mM potassium phosphate buffer, pH 6.9). The mixture was incubated at 37 °C for 15 min. The reaction was initiated by adding 30 μL of 0.5 mM p-nitrophenyl-D-glucopyranoside (Sigma Aldrich) to the mixture. The reaction mixture was incubated for 30 min at 37 °C before being stopped with 1 mL of 1 M Na_2_CO_3_. Absorbance at 400 nm was used to determine enzymatic activity. Pure acarbose served as a positive control. The IC_50_ value is the extract concentration needed to inhibit 50% of α-glucosidase activity, calculated using the following formula:(4)$$\alpha -\mathrm{Glucosidase}\;\mathrm{inhibitory}\;\mathrm{activity}\;\left(\%\right)=\frac{\left({\mathrm{Abs}}_{\mathrm{control}}-\;{\mathrm{Abs}}_{\mathrm{sample}}\right)}{{\mathrm{Abs}}_{\mathrm{control}}}\times 100$$

### 2.8. MTT Assay for RAW 264.7 Cell Cytotoxicity

The MTT (3-(4,5-dimethylthiazol2-yl)-2,5-diphenyltetrazolium bromide) assay, described by Khummueng et al. [33], was used to assess the cell viability of RAW 264.7 cells treated with PLE. Briefly, RAW264.7 cells were seeded at a density of 3.0 × 10^5^ cells/cm^2^ in a 96-well plate and allowed to grow for 24 h in an incubator. Cells were treated for 24 h with PLE concentrations ranging from 0.98 to 1000 µg/mL. After 24 h of incubation, the culture medium was removed from each well. The 200 µL of 0.5 mg/mL MTT solution was added (Life Technologies, Thermo Fisher Scientific, Waltham, MA, USA) and incubated for 4 h. After removing the MTT solution, the formazan crystals were dissolved in 200 µL of dimethyl sulfoxide. The absorbance was then measured at 560 nm with a microplate reader, and the 670 nm background was subtracted. Untreated cells (without extract, lipopolysaccharide (LPS), and dexamethasone (DEX) addition) were employed as a control. The following formula was used to calculate the percentage of cell viability.
(5)$$\mathrm{Cell}\;\mathrm{viability}\;\left(\%\right)=\frac{A_{\mathrm{treated}}}{A_{\mathrm{untreated}}}\times 100$$

### 2.9. Determination of Nitric Oxide (NO) Inhibitory Activity

NO production was assessed by measuring the amount of nitrite in the culture medium using the Griess reagent [34]. Briefly, the cells were pre-treated with PLE at various concentrations for an hour. The samples used were identical to those used in the cytotoxicity tests. The medium was then supplemented with 50 ng/mL LPS and incubated for 24 h. After 24 h of incubation, 75 μL of supernatant from each well of cell culture plates was combined in a 96-well plate with 65 µL of distilled water and 10 µL of Griess reagent (1% sulfanilamide and 0.1% naphthylethylene in a 2.5% phosphoric acid solution). The samples were incubated at room temperature for 30 min before being measured at 540 nm with a microplate reader (Thermo Fisher Scientific). The nitrite concentration was determined using a sodium nitrite standard curve. Untreated cells (without extract, LPS, and DEX addition) were employed as a control.

### 2.10. Statistical Analysis

A completely randomized design (CRD) was used to set up the experiment. All experiments were conducted in triplicate and the data were presented as mean ± standard deviation (SD). ANOVA was used, and mean comparisons were analyzed using Duncan’s multiple range tests. A Pearson correlation analysis between phytochemical compositions and bioactivities was also performed. A *p*-value of <0.05 or <0.01, <0.001, or <0.0001, depending on the scenario, was considered statistically significant. Statistical analysis was conducted using the Statistical Package for Social Science (SPSS, 16.0 for Windows, SPSS Inc., Chicago, IL, USA).

## 3. Results and Discussion

### 3.1. Proximate Compositions of PL

Table 1 shows the proximate compositions of three PL varieties: Khaek Dam (KD), Holland (H), and Thai Local (L). The moisture, ash, protein, fat, carbohydrate, and fiber contents of PLs varied depending on the papaya varieties. For all three species, protein ranged from 25.96 to 32.18%, fat from 7.34 to 11.66%, carbohydrate from 5.80 to 17.91%, moisture from 6.02 to 6.49%, ash from 11.23 to 12.40%, and fiber from 23.24 to 38.48%. The L cultivar had significantly higher protein and carbohydrate contents (*p* < 0.05). The H cultivar had the highest ash content (*p* < 0.05). The KD cultivar had significantly greater amounts of fat and fiber contents than other cultivars (*p* < 0.05). The protein content in all three cultivars was higher than what was reported by Martial-Didier et al. [35], which was 9.82% (dry weight). 

The proper recycling of PL as byproducts in the current study could result in a variety of cost-effective protein-rich products. In this current study, the PL powder had a high dietary fiber content (23.24–38.48%, dry weight). This finding indicated the importance of PL as a source of dietary fiber, based on Anderson et al. [36], in which the authors define high fiber foods as those with a content of more than 6%.

### 3.2. Bioactive Constituents of PLE

Total phenolic content (TPC), total flavonoid content (TFC), and tannin content (TC) of three PL extracts are shown in Figure 1. The TPC and TC ranged from 64.47 to 128.90 mg GAE/g and 113.94 to 173.69 mg GAE/g, respectively (Figure 1). The highest TPC and TC were found in the PL extract of the H cultivar (173.69 mg GAE/g extracts and 128.90 mg GAE/g), which was followed by the KD cultivar (113.94 mg GAE/g and 64.47 mg GAE/g) and the L cultivar (156.50 mg GAE/g and 107.00 mg GAE/g). Nandini et al. [37] reported that an ethanolic extract of PLs from Mysore, India, had a TPC of 13.5 mg GAE/g, which was lower than the TPC found in all three PL cultivars in this study. Similarly, TPC of aqueous PLE from India was 14.53 mg GAE/g, while TPC of Indonesian papaya cultivars ranged from 119.84 to 124.18 mg GAE/g [38]. Rahayu et al. [39] reported that the TPC of PLs in Thailand ranged from 105.15 to 105.92 mg GAE/g, while Chaithada et al. [40] found that the TPC of ethanolic extracts from three cultivars of PL (Holland, Khak Dam, and Red Lady) obtained from Surat Thani and Phatthalung, Thailand, were 13.38 mg GAE/g, 84.99 mg GAE/g, and 169.85 mg GAE/g, respectively. Furthermore, Gaye et al. [41] reported that the TPC of PLs from common cultivars, Red Lady, and Sunrise in Senegal were 29.75 mg GAE/g, 53.24 mg GAE/g, and 41.77 mg GAE/g, respectively. The TPC of ethanolic extracts of PL from Pakistan was 65.12 mg GAE/g [42], whereas Malaysian PL had 102.59 mg GAE/g [43]. These findings suggested that the TPC of PLs varies depending on the cultivar and cultivated location.

The total flavonoid content (TFC) varied from 41.35 to 87.63 mg RE/g (Figure 1). The KD extract had the highest TFC at 87.63 mg RE/g (*p* < 0.05), followed by the L extract (64.43 mg RE/g) and the H extract (41.35 mg RE/g). The TFC in all PLEs were higher than those reported by Chaithada et al. [40] for three PL cultivars in Thailand: Holland (69.88 µg QE/g), Khak Dam (155.45 µg QE/g), and Red Lady (276.72 µg QE/g). In line with Gaye et al. [41], the TFC of Senegalese old-leaf ordinary common papaya (1.11 mg QE/g), Red Lady (1.46 mg QE/g), and sunrise (1.39 mg QE/g) was lower than that of our current study (Figure 1). TFC in Indonesian papaya cultivars varied from 36.93 to 76.69 µg QE/g [17]. However, the TFC of the three PLEs used in this study was lower than that reported by Rahayu et al. [39] for Indonesian PL (340–350 mg QE/g) and Thai PL (315–540 mg QE/g). Flavonoids are a major class of phenolic compounds used in traditional medicine. Flavonoid is a secondary metabolite product that acts as an antioxidant by scavenging free radicals because they are potential reducing agents and contribute to the extract’s oxidative properties [44]. Flavonoids are also capable of treating certain physiological disorders and diseases; they sometimes appear as glycosides and have several phenolic hydroxyl groups on their ring structure. Some flavonoids have been shown to have a variety of biological activities, including antimicrobial, anti-inflammatory, antiangiogenesis, antidiabetic, analgesic, antiallergic, and cytostatic properties [44]. 

TC in the three PLEs ranged from 66 to 128 mg GAE/g extract, with H-PLE having the highest tannin content (*p* < 0.05, Figure 1). It should be noted that H-PLE had higher tannin content than flavonoid content (*p* < 0.05), while there was no significant difference between tannin and flavonoid contents in L-PLE (*p* > 0.05). This result was lower than the levels of tannin in the PLE reported by Ugo et al. [45], which were 310.50 mg/100 g. Tannins have antimicrobial properties due to iron deprivation, hydrogen binding, or specific interactions with vital proteins such as enzymes in microbial cells, and they have potential in cancer prevention [46]. Tannins were shown by Li et al. [47] to be useful in the treatment of inflamed or ulcerated tissues.

According to the results and previous reports, the papaya cultivar, growing location, solvent extraction, and extraction methods all had an impact on TPC, TFC, and tannin content.

### 3.3. Phenolic and Metabolic Profiles of PLE

The phenolic and metabolic profiles of the three PL cultivars were identified using liquid chromatography-mass spectrometry analyses (LC/MS) (Table 2 and Table 3). A total of 34 phenolic compounds were identified in all three PL cultivars, including simple phenols, phenolic acids and their derivatives, flavonols, tannins, terpenoid, glycosides, and indole and its derivatives (Table 2). All three PL cultivars had similar phenolic profiles, with ferulic acid, kaempferol, and quercetin being the most abundant polyphenols. The phenolic profiles of KD, H, and L cultivars were like previous studies by Nugroho et al. [48] in PLs from Indonesia; flavonoids identified include quercetin, kaempferol 3-rutinoside, quercetin 3-(2G rhamnosylrutinoside), and quercetin 3-rutinoside. The Malaysian PL also contained kaempferol, ferulic acid, caffeic acid, myricetin, carpaine, pseudocarpaine, dehydrocarpaine I and II, chlorogenic acid, β-carotene, lycopene, and anthraquinonesglycoside [13]. The Uganda PL contained phenolic acids (caffeic acid, *p*-coumaric acid, and ferulic acid) as well as flavonoids (quercetin dirhamnosyl-hexoside, kaempferol dirhamnosyl-hexoside, and quercetin 3-*O*-rutinoside) [12].

The other metabolic profiles of three PL cultivars are summarized in Table 3, which includes 26 amino acids and derivatives, 18 lipids and derivatives, 8 carbohydrates and derivatives, and 38 organic compounds. Metabolic compounds are precursors to the synthesis of secondary metabolites, which are essential for plant development and stress response [25]. Jadaun et al. [49] observed a variety of metabolic compounds in Indian PL, including steroids, alkaloids (peptides and amino acids), glycosides, lipids, phenolic compounds (aromatic phenol, quinone, and flavonoids), terpenes, aliphatic compounds (fatty acids, alcohol, saturated, and unsaturated alkenes), and other bioactive compounds. Moreover, Indian papaya seed contained 14 metabolic compounds, including oleic acid, stearic acid, methyl ester, dibenzoyl-L-tartaric acid, palmitic acid, and phenol, 2-methoxy-5-prophenyl-(E) [49]. Ghosh et al. [50] also identified oleic acid as the main compound in PLs. They also found certain secondary metabolites in PLs such as caffeine, cinnamic, chlorogenic, quinic, *p*-coumaric, vanillic, and protocatechuic acids, as well as naringenin, hesperidin, rutin, and kaempferol. In addition, PLs were composed of organic acids (lactic, quinic, propionic, succinic, citric, malic, and fumaric acids), lipids (palmitic acid and linolenic acid), and carbohydrates (raffinose, beta glucose, alpha glucose, and sucrose) [51].

Chromatograms in positive and negative modes, and PCA in positive and negative modes of metabolite profiles of three PLE acquired via LC-HRMS/MS are shown in Figure 2.

### 3.4. Antioxidant Activities of PLE

Table 4 displays the IC_50_ values for DPPH^•^ scavenging activity, metal chelation ability, and reducing power in three PLEs. The KD-PLE had the lowest IC_50_ value of DPPH^•^ scavenging activity at 4.08 mg/mL (Table 4, *p* < 0.05), making it the most effective free-radical scavenger. This was in accordance with the findings of Chaithada et al. [40], who observed that the KD-PLE had the highest DPPH^•^ scavenging activity. The IC_50_ values for L- and H-PLE were 5.14 mg/mL and 5.92 mg/mL. All three PLEs had significantly higher IC_50_ values for DPPH^•^ scavenging activity compared to standard α-tocopherol (32.27 µg/mL), indicating lower radical-scavenging activity. According to Soib et al. [13], Eksotika PLE from Malaysia recovered via reflux extraction had the highest DPPH^•^ scavenging activity with an IC_50_ value of 0.24 mg/mL, followed by ultrasonic-assisted extraction (UAE) (IC_50_ = 0.38 mg/mL) and agitation (IC_50_ = 0.40 mg/mL). Yap et al. [52] observed that PLE from Malaysia had significantly higher (*p* < 0.05) DPPH^•^ scavenging activity (IC_50_ = 298 µg/mL) than stalks (IC_50_ = 1619 µg/mL). The antioxidant activity of PL was attributed to the presence of carpaine, its primary active compound [52].

The PLE of the KD and L cultivars had the lowest IC_50_ value for metal chelating activity (IC_50_ = 0.31 mg/mL; *p* < 0.05), indicating the greatest metal chelating ability. It should be noted that PLE of both cultivars demonstrated significantly lower metal chelating activity than EDTA (20.07 µg/mL). Metal chelating capacity, particularly for iron, refers to a compound’s ability to bind metal ions by preventing metal ions from producing reactive oxygen species (ROS), thereby reducing metal-induced oxidative stress and protecting against oxidative damage [53].

Table 4 shows the reducing activity of three different PLEs as measured with the potassium ferrocyanide reduction method. The reducing power of three PLEs differed significantly (*p* < 0.05), with the H extract having the highest reducing power (19.14 mg AAE/g extract), followed by the KD extract (15.28 mg AAE/g extract) and the L extract. This implied that the H extract possessed superior electron and/or hydrogen donors. The higher TPC of the H extract was responsible for its greater reducing power (Figure 1). Irondi et al. [22] found that PLE from Nigeria had a reducing power of 6.82 mg GAE/g, while papaya fruit extracts from Ethiopia had a reducing power value of 13.5 mg AAE/g [54]. Omar et al. [55] reported that Malaysian papaya seeds had a ferric reducing power range of 5.718 to 11.758 mg GAE/g.

The KD extract had lower TPC than the H and L extracts (Figure 1), but it contained more TFC. This could account for the higher DPPH^•^ scavenging activity, metal chelating activity, and reducing power of the KD extract, which were primarily due to the flavonoids present. The antioxidant activity of the PLEs were generally consistent with their total phenolic and flavonoid concentrations. Phytochemicals, particularly plant phenolics like flavonoids and phenolics, are a significant group of compounds that function as primary antioxidants [56]. Polyphenols and flavonoids with specific hydroxyl positions can act as proton donors and exhibit radical-scavenging activity [56]. These compounds had antioxidant activity because they neutralized lipid free radicals and prevented hydroperoxide breakdown into free radicals [57].

### 3.5. α-Amylase and α-Glucosidase Inhibitory Activity

Table 4 shows the inhibitory activities of three PLEs against α-amylase and α-glucosidase, which are associated with glycemic control associated with antidiabetes properties. Enzymes like α-amylase break down complex polysaccharides into oligosaccharides and disaccharides. α-glucosidase converts oligosaccharides and disaccharides to monosaccharides, which are then absorbed into the hepatic portal vein via enterocytes in the small intestine. Typically, postprandial glucose levels increase in response to high-carbohydrate diets [58]. Inhibiting α-amylase and α-glucosidase may slow glucose absorption into the bloodstream, resulting in lower postprandial blood glucose levels [58]. This glycemic control can be especially useful for patients with type 2 diabetes [58]. The KD extract had the lowest IC_50_ value of 2.28 mg/mL against α-amylase, while the H and L extracts showed moderate activity toward α-amylase (IC_50_ = 2.36 mg/mL and 2.69 mg/mL, respectively). The KD extract (IC_50_ = 1.73 mg/mL) had a lower IC_50_ value against α-glucosidase than the H and L extracts (IC_50_ = 2.56 mg/mL and 4.20 mg/mL, respectively). It should be noted that the KD extract showed significant inhibitory activity against α-amylase and α-glucosidase. However, all three PLE cultivars had higher IC_50_ values than standard acarbose, indicating lower antidiabetic activity. The antidiabetic effect of PLE may be due to the synergistic action of compounds found in the leaf, such as phenolic glycosides. These compounds can inhibit the enzyme α-glucosidase by acting as modulating substrates [59]. Phenolic compounds, such as flavonoids and tannins, can help prevent and manage diabetes [60]. According to Table 2, the KD extract contained more glycosides than the H and L extracts. Different phenolic profiles may have different inhibitory activity against α-amylase and α-glucosidase. PLE could potentially reduce carbohydrate hydrolyzing enzyme activity in the small intestines by binding to active sites of α-amylase and α-glucosidase. This then inhibits enzyme action and subsequently slows down the breakdown of complex carbohydrates into glucose molecules [61].

### 3.6. Correlation Analysis between Phytochemical Compositions and Bioactivities

Table 5 shows the Pearson’s correlation coefficients for different phytochemical compositions and PLE bioactivities. TPC, TFC, TC, and selected metabolites were evaluated depending on their intensity in each group. Examples included hydroxycinnamic acid (ferulic acid), flavonols (kaempferol and quercetin), and glycosides (caffeic acid 3-glucoside). The DPPH free-radical scavenging activity, metal chelation, α-amylase, and α-glucosidase inhibitory activities were expressed as IC_50_ values, with the lower value indicating higher activity. The reducing power was shown as mg AAE/g, with the higher value signifying greater activity. The results showed that the bioactivity of PLEs was determined by particular components rather than the total concentration of phenolic, flavonoid, and tannin. DPPH free-radical inhibition demonstrated significant correlations with TFC (*p* < 0.01) and caffeic acid 3-glucoside (*p* < 0.05). It is widely acknowledged that flavonoids exhibit substantial free-radical scavenging activity, as Tsimogiannis and Oreopoulou [62] showed the role of flavonoid structure to DPPH free-radical scavenging efficiency.

Reducing power significantly correlated with the TC (*p* < 0.05). Crude tannins from numerous plant species, including canola and rapeseed hulls, amaranth (*Amaranthus caudatus* L.), and pomegranate juice, peel, and seed (*Punica granatum* L.), have been shown to exhibit reducing power [63,64,65]. Tannins in the leaves, twigs, and stem bark of *Canarium album* or Chinese olive (Burseraceae) have been shown to increase ferric reducing power as concentration increases [66]. Metal chelation showed a strong correlation with TFC (*p* < 0.01), kaempferol (*p* < 0.01), and quercetin (*p* < 0.05). Flavonols, including kaempferol and quercetin, have been discovered to exhibit substantial metal ion chelation when investigated using UV spectroscopy and electrospray ionization [67]. There was a considerable correlation between α-amylase and α-glucosidase inhibitory activities and the presence of glycoside, namely caffeic acid 3-glucoside (*p* < 0.01). Molecular docking has shown that glycosides, such as isorhamnetin-3-*O*-glucoside, can efficiently inhibit α-glucosidase by interacting with crucial amino acids [68]. 

### 3.7. Cytotoxicity of PLE

Figure 3 depicts the effect of three PLEs on the viability of RAW 264.7 macrophage cells activated with LPS. The cell viability was determined with the concentration of PLE, and it decreased as the PLE concentration increased. Cell viability of KD-PLE (Figure 3a) and L-PLE (Figure 3c) was over 80% at doses up to 3.93 μg/mL. H extract treatment resulted in over 80% cell viability at doses up to 500 μg/mL (Figure 3b). Hyun et al. [69] observed that RAW 264.7 cells treated with PL aqueous extract at doses ranging from 12.5 µg/mL to 400 µg/mL showed no toxicity after 24 h. Common berry phenolic compounds, at the 16–500 μM range, have been shown to inhibit NO production by >50% without exhibiting cytotoxicity in RAW 264.7 macrophages. These compounds include the flavonols quercetin and myricetin, the isoflavone daidzein, and the anthocyanins/anthocyanidins pelargonidin, cyanidin, delphinidin, peonidin, malvidin, malvidin 3-glucoside, and malvidin 3,5-diglucosides [70]. However, according to Yuliani et al. [71], saponins, triterpenoids, flavonoids, and glycosides in PL contributed to its cytotoxic effect on LLC-MK2 cells. To completely comprehend the toxic effects of the PLE, it may be necessary to compare several test models, including both normal and abnormal cells, for the purpose of determining cytotoxicity in the future.

### 3.8. Nitric Oxide (NO) Inhibitory Activity of PLE

NO is a signaling molecule that plays an important role in the pathogenesis associated with inflammation [72]. The ability of PLE to inhibit NO production in LPS-stimulated RAW 264.7 cells was tested using a Griess reagent to assess its anti-inflammatory activity [73]. As shown in Figure 4, RAW 264.7 cells activated with LPS produced significantly higher NO levels than unstimulated cells. At extract doses ranging from 0.98 to 1000 μg/mL, PLE significantly reduced NO secretion in RAW 264.7 cells compared to LPS-induced cells (*p* < 0.05), indicating anti-inflammatory effects. NO is usually an intercellular mediator produced excessively by inducible nitric oxide synthase (iNOS), which is activated by bacterial products and cytokines [74]. The bioactive extracts or compounds in PLE may act by inhibiting the activity of iNOS or by having the ability to neutralize free radicals [75], thereby suppressing NO formation. The anti-inflammatory activity of PLE could be attributed to the presence of phenolic compounds and flavonoids in the extract, both of which have high antioxidant activity. Flavonoids are mostly able to inhibit NO production and reduce iNOS protein expression [73]. The presence of a planar ring structure in flavonoid molecules is crucial for NO inhibitory activity. Typically, genipin, carnosol, ginkgolides, and β-carotene can reduce iNOS protein expression, inhibiting NO formation in LPS-stimulated RAW 264.7 cells [76]. Like PLE’s cytotoxic activity, the phytochemical constituent in the extract may alter its anti-inflammatory activity (Figure 4).

Flavonoids and phenolic acids have strong free-radical scavenging activity by regulating the activity of enzymes involved in reactive oxygen species scavenging, causing cell cycle arrest, inducing apoptosis and autophagy, inhibiting angiogenesis, and inhibiting cancer cell proliferation and metastasis [77]. PLE has been shown to interact with a wide range of molecular targets with therapeutic potential to counter a number of diseases. Reducing the activity of DNA topoisomerase I/II, modifying signaling pathways, and downregulating or upregulating gene expression are all important biological targets in cancer prevention via PLEs [78]. Hyun et al. [69] observed that a PL aqueous extract inhibited the production of NO, prostaglandin E2 (PGE2), and pro-inflammatory cytokines by increasing iNOS and cyclo-oxygenase-2 activity, indicating immunomodulatory effects.

## 4. Conclusions

This study showed that the cultivars of *Carica papaya* leaf extract have important effects on their functional quality and bioactivity. Khaek Dam papaya leaf extract contained higher amounts of the active compounds and exhibited the strongest antioxidant capacities and glycemic control bioactivity when compared to Holland and Thai Local cultivars. The most abundant active compounds in the three papaya leaf cultivars were ferulic acid, kaempferol, and quercetin. Interestingly, all three papaya leaf cultivars had anti-inflammatory effect by reducing nitric oxide production while remaining non-toxic to RAW264.7 cells. Thus, papaya leaf extract from the Khaek Dam cultivar could be used as a promising option as a source of functional food ingredients and advanced towards applications in future studies.

## Figures and Tables

**Figure 1 foods-13-01692-f001:**
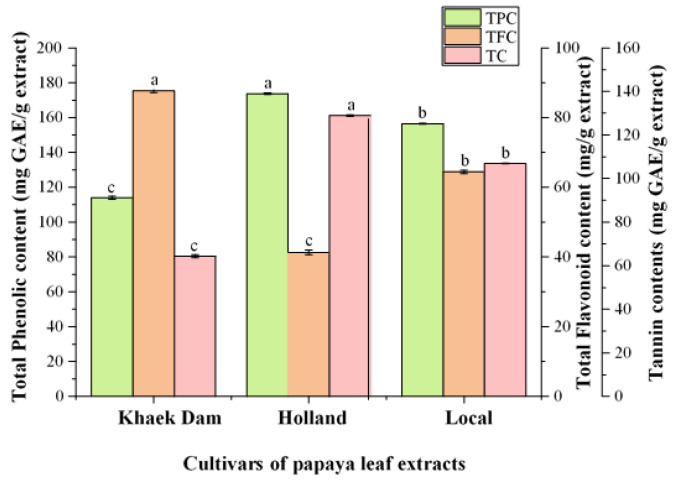
Total phenolic content (TPC), total flavonoid content (TFC), and tannin content (TC) of various cultivars of papaya leaf extracts. The standard deviations from triplicate determinations are represented by the bars. Significant differences (*p* < 0.05) are denoted by different letters.

**Figure 2 foods-13-01692-f002:**
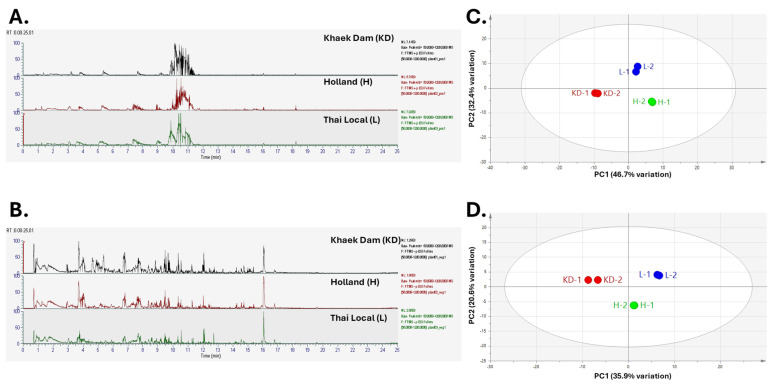
Metabolite profiles of three different varieties of papaya acquired via LC-HRMS/MS. Chromatograms in (**A**) positive and (**B**) negative modes, and PCA in (**C**) positive and (**D**) negative modes.

**Figure 3 foods-13-01692-f003:**
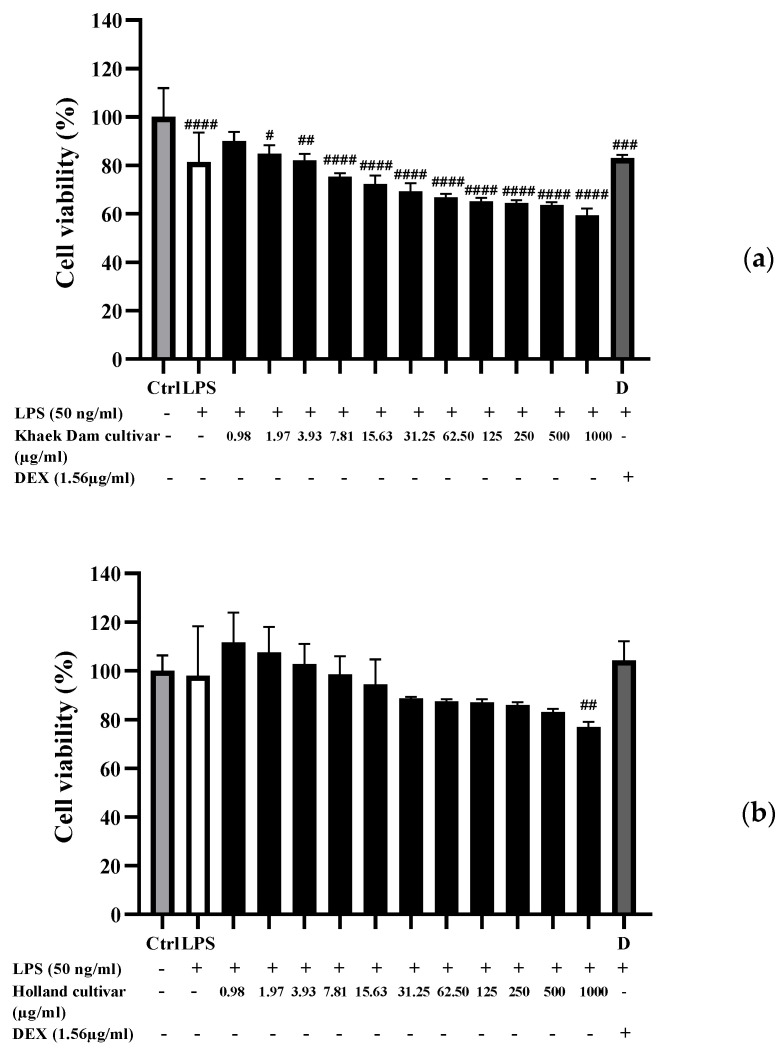
The percentage of cell viability: (**a**) Khaek Dam (KD), (**b**) Holland (H), and (**c**) Local (L) papaya leaf extracts against LPS-activated RAW264.7 macrophages. Each bar graph represents the means ± standard deviation. The ^#^, ^##^, ^###^, and ^####^ symbols indicate significant differences at *p* < 0.05, 0.01, 0.001, and 0.0001 as compared to the untreated cells (control).

**Figure 4 foods-13-01692-f004:**
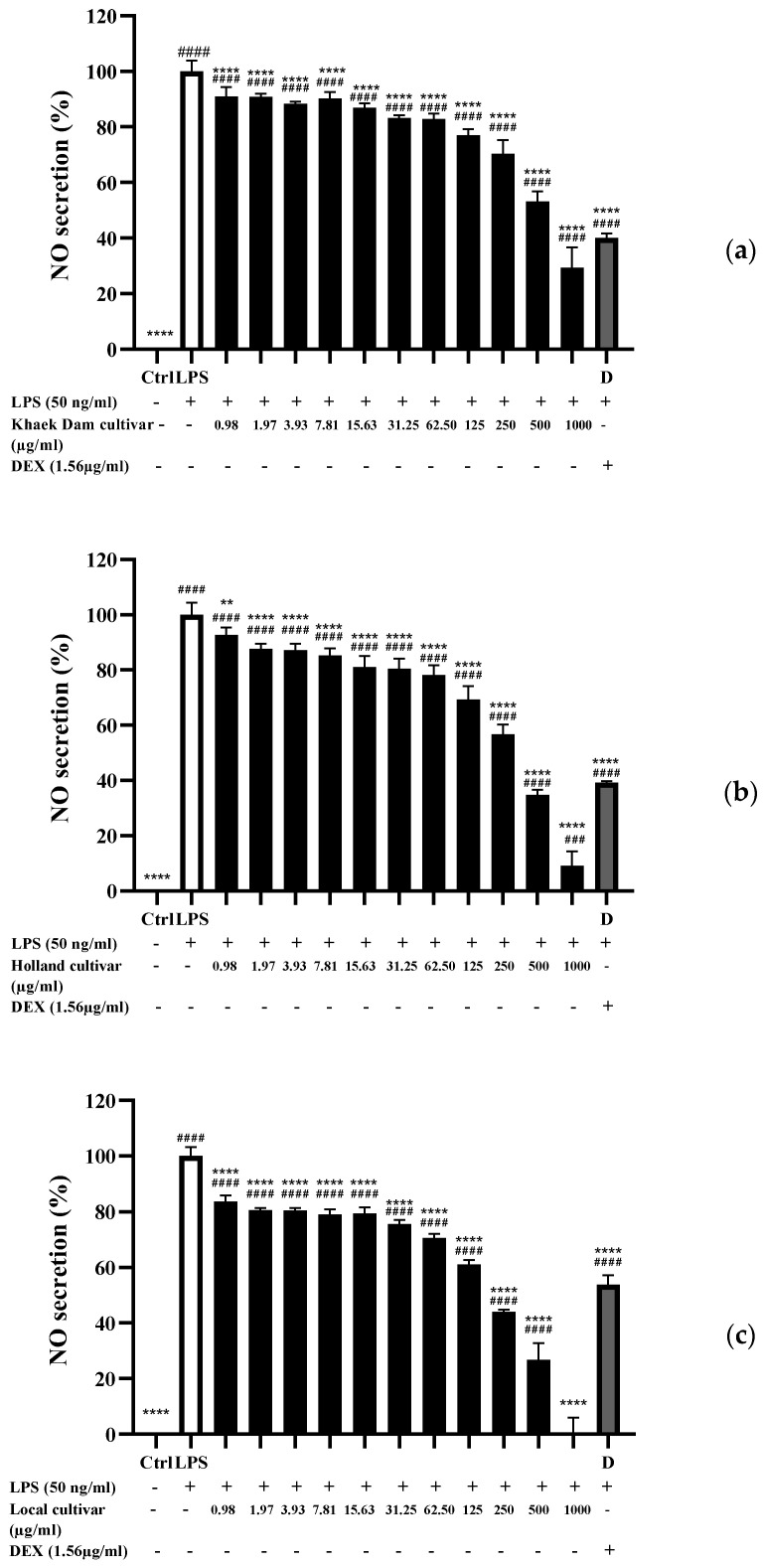
Nitric oxide (NO) inhibitory activity: (**a**) Khaek Dam (KD), (**b**) Holland (H), and (**c**) Local (L) papaya leaf extracts against LPS-activated RAW264.7 macrophages. Each bar graph represents the means ± standard deviation. The ** and **** symbols indicate significant differences at *p* < 0.01 and 0.0001 as compared to the LPS-treated cells (LPS), whereas the ^###^ and ^####^ symbols indicate significant differences at *p* < 0.001 and 0.0001 as compared to the untreated cells (control).

**Table 1 foods-13-01692-t001:** The chemical composition of three papaya leaf cultivars.

Parameter	Khaek Dam (KD)	Holland (H)	Local (L)
Moisture (%)	6.21 ± 0.05 ^b^	6.49 ± 0.04 ^a^	6.02 ± 0.01 ^c^
Ash (%)	11.63 ± 0.03 ^b^	12.40 ± 0.04 ^a^	11.23 ± 0.06 ^c^
Protein (%)	25.96 ± 0.20 ^c^	30.43 ± 0.16 ^b^	32.18 ± 0.34 ^a^
Fat (%)	11.66 ± 0.04 ^a^	7.34 ± 0.06 ^c^	9.42 ± 0.02 ^b^
Carbohydrate (%)	6.06 ± 0.19 ^b^	5.80 ± 0.11 ^b^	17.91 ± 0.36 ^a^
Fiber (%)	38.48 ± 0.04 ^a^	37.55 ± 0.30 ^b^	23.24 ± 0.36 ^c^

Values are reported as mean ± standard deviation from triplicate determinations. Different letters in the same row indicate significant differences (*p* < 0.05).

**Table 2 foods-13-01692-t002:** Data of phenolic constituents and their derivatives identified based on liquid chromatography–mass spectrometry analyses (LC/MS) in three different papaya leaf cultivars.

Retention Time (min)	Calc. MW	*m/z*	Annot. DeltaMass [ppm]	Reference Ion	Putative Metabolite	Chemical Formula	Compound Class	ChemSpider (CSID)	mzCloud Match (%)	Metabolite Identification Level *	Khaek Dam (KD)	Holland (H)	Thai Local (L)
3.006	156.06932	157.07659	3.63	[M + H] + 1	3-Indoleacetonitrile	C_10_H_8_N_2_	indole derivative	312,357	84.5	2	3.09 × 10^7^	2.48 × 10^7^	3.18 × 10^7^
3.014	203.05817	204.06543	−0.38	[M + H] + 1	Indole-3-pyruvic acid	C_11_H_9_NO_3_	indole	781	51.5	2	5.28 × 10^6^	3.01 × 10^6^	5.39 × 10^6^
4.394	164.04776	165.05504	2.55	[M + H] + 1	(E)-*p*-Coumaric acid	C_9_H_8_O_3_	hydroxycinnamic acid	553,148	n/a	2	2.45 × 10^7^	4.13 × 10^7^	3.82 × 10^7^
4.993	205.07384	188.07056	−0.24	[M + H − H_2_O] + 1	Indole-3-lactic acid	C_11_H_11_NO_3_	indole derivative	83,867	75.4	2	2.69 × 10^6^	3.12 × 10^6^	5.57 × 10^6^
5.162	198.05277	199.06005	−0.29	[M + H] + 1	Syringic acid	C_9_H_10_O_5_	hydroxybenzoic acid	10,289	85.1	2	1.26 × 10^7^	7.96 × 10^6^	4.59 × 10^6^
5.19	198.0536	199.06088	3.93	[M + H] + 1	Vanillylmandelic acid	C_9_H_10_O_5_	methoxyphenols	1207	n/a	3	1.39 × 10^7^	8.27 × 10^6^	6.85 × 10^6^
6.222	133.05266	132.04538	0.75	[M + H] + 1	5-Indolol	C_8_H_7_NO	indole	15,244	n/a	3	1.74 × 10^7^	5.58 × 10^8^	7.71 × 10^6^
7.168	216.0898	217.09708	−0.36	[M + H] + 1	2,3,4,9-Tetrahydro-1H-β-carboline-3-carboxylic acid	C_12_H_12_N_2_O_2_	indole derivative	88,749	98.5	2	2.02 × 10^7^	3.72 × 10^7^	3.21 × 10^7^
7.251	122.0373	123.04458	4.29	[M + H] + 1	Benzoic acid	C_7_H_6_O_2_	aromatic carboxylic acid	238	n/a	3	2.84 × 10^6^	5.23 × 10^6^	2.75 × 10^6^
7.89	154.1363	155.14357	3.44	[M + H] + 1	(+/−)-Eucalyptol	C_10_H_18_O	terpenoid	2656	n/a	3	2.99 × 10^6^	2.47 × 10^6^	9.75 × 10^6^
7.9	108.05788	109.06516	3.37	[M + H] + 1	*p*-Cresol	C_7_H_8_O	simple phenols	13,839,082	n/a	3	3.16 × 10^6^	5.84 × 10^5^	7.51 × 10^5^
8.287	124.05285	125.06013	3.4	[M + H] + 1	Guaiacol	C_7_H_8_O_2_	methoxyphenols	447	n/a	3	6.51 × 10^6^	5.36 × 10^6^	2.54 × 10^6^
8.315	152.04803	153.0553	4.48	[M + H] + 1	Vanillin	C_8_H_8_O_3_	phenolic aldehyde	13,860,434	80.8	2	4.02 × 10^7^	4.69 × 10^7^	2.66 × 10^7^
8.376	302.04257	303.04985	−0.28	[M + H] + 1	Quercetin	C_15_H_10_O_7_	flavonols	4,444,051	98.9	2	2.05 × 10^8^	1.63 × 10^8^	2.66 × 10^8^
9.372	316.05952	317.0668	3.85	[M + H] + 1	Isorhamnetin	C_16_H_12_O_7_	flavonols	96	83	2	9.65 × 10^6^	7.19 × 10^6^	1.75 × 10^7^
9.534	148.05302	149.0603	3.98	[M + H] + 1	Cinnamic acid	C_9_H_8_O_2_	aromatic carboxylic acid	392,447	n/a	3	2.55 × 10^7^	1.69 × 10^7^	1.55 × 10^7^
10.105	286.04758	287.05485	−0.56	[M + H] + 1	Kaempferol	C_15_H_10_O_6_	flavonols	4,444,395	97.3	2	2.60 × 10^8^	1.49 × 10^8^	3.81 × 10^8^
11.294	194.05788	177.0546	−0.13	[M + H − H_2_O] + 1	(E)-Isoferulic acid	C_10_H_10_O_4_	hydroxycinnamic acid	643,318	n/a	3	1.37 × 10^7^	7.09 × 10^6^	2.54 × 10^7^
12.05	164.08424	165.09152	3.12	[M + H] + 1	Eugenol	C_10_H_12_O_2_	methoxyphenols	13,876,103	n/a	3	4.55 × 10^6^	1.72 × 10^6^	1.21 × 10^6^
0.923	192.0633	191.05561	−0.48	[M − H] −1	D-(−)-Quinic acid	C_7_H_12_O_6_	cyclohexanecarboxylic acid	10,246,715	92.5	2	1.11 × 10^6^	2.49 × 10^6^	2.50 × 10^6^
3.907	154.02656	153.01929	−0.29	[M − H] − 1	Protocatechuic acid	C_7_H_6_O_4_	hydroxybenzoic acid	71	n/a	3	2.94 × 10^5^	3.49 × 10^6^	8.47 × 10^5^
4.097	332.07418	331.06687	−0.51	[M − H] − 1	Glucogallin	C_13_H_16_O_10_	tannins	110,537	n/a	3	6.88 × 10^6^	8.46 × 10^7^	3.18 × 10^7^
4.962	224.03211	223.02483	0.1	[M − H] − 1	3-[(1-Carboxyvinyl)oxy]-4-hydroxybenzoic acid	C_10_H_8_O_6_	hydroxybenzoic acid	8,096,552	n/a	3	5.60 × 10^5^	7.26 × 10^5^	3.40 × 10^6^
5.653	312.0485	293.03068	1.17	[M − H − H_2_O] − 1	Caftaric acid	C_13_H_12_O_9_	hydroxycinnamic acid	4,944,664	n/a	3	1.17 × 10^6^	7.56 × 10^6^	6.23 × 10^6^
7.138	316.11503	315.10775	2.43	[M − H] − 1	Vanilloloside	C_14_H_20_O_8_	glycosides	24,695,215	n/a	3	7.39 × 10^7^	1.08 × 10^8^	3.37 × 10^7^
8.18	342.0953	341.08803	0.65	[M − H] − 1	Caffeic acid 3-glucoside	C_15_H_18_O_9_	glycosides	4,445,073	n/a	3	1.04 × 10^8^	7.46 × 10^7^	6.60 × 10^7^
8.881	94.04194	93.03467	0.85	[M − H] − 1	Phenol	C_6_H_6_O	simple phenols	971	92.7	2	7.67 × 10^6^	1.07 × 10^7^	8.13 × 10^6^
9.148	182.05809	181.05081	0.99	[M − H] − 1	Homovanillic acid	C_9_H_10_O_4_	methoxyphenols	1675	n/a	3	4.24 × 10^5^	3.48 × 10^5^	4.11 × 10^5^
9.486	194.05783	193.05055	−0.41	[M − H] − 1	Ferulic acid	C_10_H_10_ O_4_	hydroxycinnamic acid	393,368	97.1	2	2.32 × 10^8^	2.38 × 10^8^	2.49 × 10^8^
9.657	224.06841	225.07565	−0.27	[M − H] − 1	Sinapic acid	C_11_H_12_O_5_	hydroxycinnamic acid	553,361	90.4	2	1.29 × 10^7^	4.46 × 10^7^	1.22 × 10^8^
9.668	340.07935	339.07207	−0.24	[M − H] − 1	Aesculin	C_15_H_16_O_9_	glycosides	4,444,765	n/a	3	7.82 × 10^6^	6.74 × 10^6^	8.54 × 10^7^
10.166	360.14088	359.13491	0.44	[M − H] − 1	8-Epideoxyloganic acid	C_16_H_24_O_9_	glycosides	391,568	n/a	3	1.93 × 10^7^	1.49 × 10^7^	1.39 × 10^7^
10.574	138.03179	137.02451	0.66	[M − H] − 1	Salicylic acid	C_7_H_6_O_3_	hydroxybenzoic acid	331	99	2	3.26 × 10^7^	4.84 × 10^7^	2.79 × 10^7^
15.237	126.03179	125.02452	0.78	[M − H] − 1	Pyrogallol	C_6_H_6_O_3_	dihydroxyphenols	13,835,557	n/a	3	6.20 × 10^6^	4.01 × 10^6^	9.28 × 10^5^

* Metabolite identification level was followed from the Metabolomics Standards Initiative, namely level1: confidently identified compounds, level2: putatively annotated compounds, level3: putatively annotated compound classes, and level4: unknown compounds. n/a = not applicable. Different color levels in the same column represent the amount of compounds; 
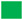
 the most, 
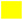
 the middle, and 

 the least.

**Table 3 foods-13-01692-t003:** Metabolites present in the three papaya leaf cultivars (Khaek Dam (KD), Holland (H) and Local (L) analyzed with liquid chromatography-mass spectrometry.

Metabolite	Compounds
Amino acid and its derivative	L-Isoleucine, L-Phenylalanine, L-Valine, D-(+)-Proline, L-(−)-Methionine, D-(+)-Tryptophan, L-Histidine, Asparagine, DL-Glutamine, DL-Arginine, 2-Aminobutyric acid, 4-Acetamidobutanoic acid, Phenylacetaldehyde, 4-Guanidinobutyric acid, L-Glutathione, N-Acetylornithine, Acetylarginine, N6,N6,N6-Trimethyl-L-lysine, Glycine, L-(+)-Alanine, L-(−)-Serine, L-(−)-Threonine, Aminolevulinic acid, Asparagine, DL-Glutamic acid, L-Tyrosine
Lipid and fatty acids and their derivatives	(R)-3-Hydroxy myristic acid, α-Linolenic acid, 9-Oxo-ODE, (±)12(13)-DiHOME, 16-Hydroxyhexadecanoic acid, Corchorifatty acid F, Hexadecanamide, Oleoyl ethanolamide, α-Linolenoyl ethanolamide, Erucamide, 5α-Dihydrotestosterone, Butyl palmitate, Stearidonic acid, (+/-)9-HODE, 13(S)-HpOTrE, 3-oxopalmitic acid, (±)9-HpODE, 2-Hydroxy-4-(methylthio)butanoic acid
Saccharide and its derivatives	D-(+)-Glucose, D-Xylonic acid, D-(−)-Fructose, D-(+)-arabitol, Gluconic acid, D-(+)-Galactose, D-(−)-Mannitol, L-Iditol
Organic compound	Caffeic acid, Citric acid, Guvacine, 4-Methoxycinnamaldehyde, DL-Malic acid, Myristicin, Fumaric acid, Glutaric acid, 4-Acetyl-2-prenylphenol, Isoamylamine, 2-Ethyl-2-phenylmalonamide, N-Phenylacetylglutamine, L-(+)-Lactic acid, Methylmalonic acid, Caprolactam, N-Acetylputrescine, Picolinic acid, 4-Hydroxybenzaldehyde, 4-Pyridoxic acid, Acetamide, Malondialdehyde, 2-Pyrrolidone, 2-Furoic acid, Maleic acid, Malonic acid, Maleamic acid, Levulinic acid, Nicotinamide, Nicotinic acid, 2-morpholinoacetic acid, Adipic acid, Safrole, Hippuric acid, Azelaic acid, (+/−)-Camphoric acid, Porphobilinogen, Indican, 5′-S-Methyl-5′-thioadenosine

**Table 4 foods-13-01692-t004:** In vitro antioxidant and antidiabetic activities of three PLEs.

Parameter	Khaek Dam (KD)	Holland (H)	Local (L)
DPPH assay ^1^ (IC_50_, mg/mL)	4.08 ± 0.004 ^c^	5.92 ± 0.009 ^a^	5.14 ± 0.017 ^b^
Reducing power (mg AAE/g)	15.28 ± 0.02 ^b^	19.14 ± 0.04 ^a^	14.66 ± 0.04 ^c^
Metal chelation ^2^ (IC_50_, mg/mL)	0.31 ± 0.001 ^b^	0.33 ± 0.001 ^a^	0.31 ± 0.001 ^b^
α-Amylase inhibitory activity ^3^ (IC_50_, mg/mL)	2.28 ± 0.02 ^c^	2.36 ± 0.01 ^b^	2.69 ± 0.03 ^a^
α-Glucosidase inhibitory activity ^4^ (IC_50_, mg/mL)	1.73 ± 0.01 ^c^	2.56 ± 0.02 ^b^	4.20 ± 0.05 ^a^

Values are reported as mean ± standard deviation from triplicate determinations. Different letters in the same row indicate significant differences (*p* < 0.05). ^1^ IC_50_ value of DPPH^•^ scavenging activity of α-tocopherol was 32.27 μg/mL. ^2^ IC_50_ value of metal chelation of EDTA was 20.07 μg/mL. ^3^ IC_50_ value of α-amylase inhibitory activity of acarbose was 16.61 μg/mL. ^4^ IC_50_ value of α-glucosidase inhibitory activity of acarbose was 14.99 μg/mL.

**Table 5 foods-13-01692-t005:** Pearson’s correlation coefficients between some phytochemical compositions and bioactivities of PLE.

	TPC	TFC	TC	Ferulic Acid	Kaempferol	Quercetin	Caffeic Acid 3-Glucoside
DPPH assay (IC_50_, mg/mL)	0.988 **	−0.996 **	0.995 **	0.429	−0.400	−0.324	−0.794 *
Reducing power (mg AAE/g)	0.628	−0.794 *	0.672 *	−0.292	−0.913 **	−0.877 **	−0.177
Metal chelation (IC_50_, mg/mL)	0.840 **	−0.942 **	0.869 **	0.024	−0.736 **	−0.679 *	−0.4778
α-Amylase inhibitory activity (IC_50_, mg/mL)	0.413	−0.189	0.361	0.981 **	0.768 *	0.817 **	−0.798 **
α-Glucosidase inhibitory activity (IC_50_, mg/mL)	0.545	−0.332	0.496	1.000 **	0.671 *	0.729 *	−0.881 **

TPC = total phenolic content, TFC = total flavonoid content, and TC = tannin content. * Correlation is significant at the 0.05 level (2-tailed). ** Correlation is significant at the 0.01 level (2-tailed).

## Data Availability

The original contributions presented in the study are included in the article, further inquiries can be directed to the corresponding author.

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
