# Peer review of "Phenolic and Metabolic Profiles, Antioxidant Activities, Glycemic Control, and Anti-Inflammatory Activity of Three Thai Papaya Cultivar Leaves"

_foods, 2024, doi:10.3390/foods13111692_

Round 1
Reviewer 1 Report
Comments and Suggestions for Authors
This manuscript tries to present the potential of papaya leaf extract from three different cultivars, namely Khaek Dam 24 (KD), Holland (H), and Thai Local (L). The leaf extract from these cultivars was assessed for its bioactive composition, in vitro biological activities, as well as the proximate composition of the leaves. This research carries out many detailed analyses, Yet several things need to be clarified and revised accordingly.
1. Please provide the complete binomial name (Carica papaya L.) on line 43.
2. Citation format: Revise the citation format throughout the manuscript. When authors cite a source with two or more authors, it should be written as et al. For example, on line 73: "In the study by Soib et al. [13],..." Similarly, correct the citation format in lines 86, 263, 268, 318, and 321. Incorrect citation format is found in line 250 (Steel and Torrie, 1980).
3. Subsection 2.5: Describe how the normative abundance shown in Table 2 was calculated. The footnote in Table 2 should be clearly described in this methodology section.
4. Line 323: As written in the manuscript, the biological activities of PL have been influenced by cultivar, growing location, solvent extraction, and extraction methods. However, the elaboration of these statements with respect to the results shown in this study is limited. Authors should provide a more robust discussion related to these statements to clearly justify the results obtained.
5. Include the chromatogram of LC/MS results. From Table 2, indicate if any standard compounds were injected for analysis.
6. Considering the biological activities of PL from three different cultivars are influenced by the presence of phenolic compounds, a correlation analysis should be conducted in this research to provide the relationship between the tested activities.
7. When evaluating cytotoxicity activity, it is important to consider the effects of PL extract on both normal and abnormal cells. Did the authors conduct the cytotoxicity activity in both types of cells? In the manuscript, only information on cytotoxicity against RAW264.7 cells is provided.
8. Please provide more detailed information regarding how the cytotoxicity assay was conducted, including the controls used. Information related to the analysis in the results section must be described in the methodology. Discussion on cytotoxicity of PL must be added in the discussion section. More literature search is needed.
9. Before suggesting that PL extract could be used as functional food ingredients, it should be noted that cytotoxicity testing against normal cell lines must be conducted to evaluate its safety. Therefore, without this information, the suggestion for functional food ingredients in the conclusion must be deleted.
Author Response
Reviewer#1
This manuscript tries to present the potential of papaya leaf extract from three different cultivars, namely Khaek Dam 24 (KD), Holland (H), and Thai Local (L). The leaf extract from these cultivars was assessed for its bioactive composition, in vitro biological activities, as well as the proximate composition of the leaves. This research carries out many detailed analyses, Yet several things need to be clarified and revised accordingly.
- Please provide the complete binomial name (Carica papaya L.) on line 43.
Ans: Done.
- Citation format: Revise the citation format throughout the manuscript. When authors cite a source with two or more authors, it should be written as et al. For example, on line 73: "In the study by Soib et al. [13],..." Similarly, correct the citation format in lines 86, 263, 268, 318, and 321. Incorrect citation format is found in line 250 (Steel and Torrie, 1980).
Ans: Thank you very much for your insightful and comprehensive review. The citation format was thoroughly checked.
- Subsection 2.5: Describe how the normative abundance shown in Table 2 was calculated. The footnote in Table 2 should be clearly described in this methodology section.
Ans: The normative abundance shown in Table 2 were described in Materials and Methods section No. 2.5. The footnote in Table 2 has been moved and revised in the section No. 2.5 already.
- Line 323: As written in the manuscript, the biological activities of PL have been influenced by cultivar, growing location, solvent extraction, and extraction methods. However, the elaboration of these statements with respect to the results shown in this study is limited. Authors should provide a more robust discussion related to these statements to clearly justify the results obtained.
Ans: Since the "Bioactive constituents of PLE" section was written based on the findings of this study as well as research previously published in the literature. To explain this point, the text has been modified to "According to the results and previous reports, the papaya cultivar, growing location, solvent extraction, and extraction methods all had an impact on TPC, TFC, and tannin content". As a result, it is reasonable to conclude that all of the aforementioned characteristics can have an impact on the bioactive constituents of PL.
- Include the chromatogram of LC/MS results. From Table 2, indicate if any standard compounds were injected for analysis.
Ans: We have included chromatograms of PL in Fig.2. For Table 2, putative metabolites were annotated based on predicted elemental composition and precursor mass. There was no standard compound used for confirmation. We have revised Table 2 by adding metabolite identification level of each putative metabolite for their confidence.
- Considering the biological activities of PL from three different cultivars are influenced by the presence of phenolic compounds, a correlation analysis should be conducted in this research to provide the relationship between the tested activities.
Ans: A Pearson correlation analysis between phytochemical compositions and bioactivities was also performed. An extensive discussion was included, along with updated references.
3.6 Correlation analysis between phytochemical compositions and bioactivities
Table 5 shows the Pearson's correlation coefficients for different phytochemical compositions and PLE bioactivities. TPC, TFC, TC, and selected metabolites were evaluated depending on their intensity in each group. Examples included phenolic compounds (ferulic acid), flavonoids (kaempferol and quercetin), and glycosides (caffeic acid 3-glucoside). The DPPH free radical scavenging activity, metal chelation, amylase, and glycosidase inhibitory activities were expressed as IC50 values, with the lower value indicating higher activity. The reducing power was shown as mg AAE/g, with the higher value signifying greater activity. The results showed that the bioactivity of PLE was determined by particular components rather than the total concentration of phenolic, flavonoid, and tannin. DPPH free radical inhibition demonstrated to correlate effectively with TFC (p < 0.01) and caffeic acid 3-glucoside (p < 0.05). It is widely acknowledged that flavonoids exhibit substantial free radical scavenging activity, as Tsimogiannis and Oreopoulou [72] showed the role of flavonoid structure to DPPH free radical scavenging efficiency.
Reducing power significantly correlated with the TC (p < 0.05). Crude tannins from numerous plant species, including canola and rapeseed hulls, amaranth (Amaranthus caudatus L.), and pomegranate juice, peel, and seed (Punica granatum L.), have been shown to show reducing power [73, 74, 75]. Tannins in the leaves, twigs, and stem bark of Canarium album or Chinese olive (Burseraceae) have been shown to increase ferric reducing power as concentration increases [76]. Metal chelation showed a strong correlation with TFC (p < 0.01), kaempferol (p < 0.01), and quercetin (p < 0.05). Flavones, including kaempferol and quercetin, have been discovered to exhibit substantial metal ion chelation when investigated using UV spectroscopy and electrospray ionization [77]. There was a considerable correlation between a-amylase and b-glucosidase inhibitory activities and the presence of glucoside, caffeic acid 3-glucoside (p < 0.01). Molecular docking has shown that glycosides, such as isorhamnetin-3-O-glucoside, can efficiently inhibit α-glucosidase by interacting with crucial amino acids [78].
- When evaluating cytotoxicity activity, it is important to consider the effects of PL extract on both normal and abnormal cells. Did the authors conduct the cytotoxicity activity in both types of cells? In the manuscript, only information on cytotoxicity against RAW264.7 cells is provided.
Ans: We sincerely appreciate your recommendation. Since the MTT test is used to measure cellular metabolic activity as an indicator of cell viability, proliferation, and cytotoxicity, it was used in the present study. For this reason, in vitro screens using the RAW264.7 murine macrophage cell line are widely performed. Therefore, as stated in the manuscript, we exclusively used RAW264.7 cells in our investigation. In addition, the subheading was updated to "MTT assay for RAW 264.7cell cytotoxicity" to indicate the type of cell employed.
The statement “To completely comprehend the toxic effects of the PLE, it may be necessary to compare several test models, including both normal and abnormal cells, for the purpose of determining cytotoxicity in the future.” was also added in the discussion in Section 3.6.
- Please provide more detailed information regarding how the cytotoxicity assay was conducted, including the controls used. Information related to the analysis in the results section must be described in the methodology. Discussion on cytotoxicity of PL must be added in the discussion section. More literature search is needed.
Ans: Details on the cytotoxicity assay determination were provided, as well as information about the control. The discussion of PLE's cytotoxicity was expanded with more references.
“Figure 2 depicts the effect of three PLE on the viability of RAW 264.7 macrophage cells activated with LPS. The cell viability was determined by the concentration of PLE, and it decreased as the PLE concentration increased. Cell viability of KD-PLE (Fig. 2a) and L-PLE (Fig. 2c) was over 80% at doses up to 3.93 μg/mL. H extract treatment resulted in over 80% cell viability at doses up to 500 μg/mL (Fig. 2b). Hyun et al. [79] observed that RAW 264.7 cells treated with PL aqueous extract at doses ranging from 12.5 µg/mL to 400 µg/mL showed no toxicity after 24 h. Common berry phenolic com-pounds, at the 16–500 μM range, have been shown to inhibit NO production by >50% without exhibiting cytotoxicity in RAW 264.7 macrophages. These compounds include the flavonols quercetin and myricetin, the isoflavone daidzein, and the anthocyanins/anthocyanidins pelargonidin, cyanidin, delphinidin, peonidin, malvidin, malvidin 3-glucoside, and malvidin 3,5-diglucosides [80]. However, according to Yuliani et al. [81], saponins, triterpenoids, flavonoids, and glycosides in PL contributed to its cytotoxic effect on LLC-MK2 cells. To completely comprehend the toxic effects of the PLE, it may be necessary to compare several test models, including both normal and abnormal cells, for the purpose of determining cytotoxicity in the future.”
- Before suggesting that PL extract could be used as functional food ingredients, it should be noted that cytotoxicity testing against normal cell lines must be conducted to evaluate its safety. Therefore, without this information, the suggestion for functional food ingredients in the conclusion must be deleted.
Ans: The conclusion was revised and ended with a recommendation for further studies. “This study showed that the cultivars of Carica papaya leaf extract have important effects on their functional quality and bioactivity. Khaek Dam papaya leaf extract contained higher amounts of the active compounds and exhibited the strongest anti-oxidant capacities and glycemic control bioactivity when compared to Holland and Thai Local cultivars. The most abundant active compounds in the three papaya leaf cultivars were ferulic acid, kaempferol, and quercetin. Interestingly, all three papaya leaf cultivars had anti-inflammatory effect by reducing nitric oxide production while remaining non-toxic to RAW264.7 cells. Thus, papaya leaf extract from the Khaek Dam cultivar could be used as a promising option as a source of alternative ingredients and advanced towards applications in future studies.”
The statement “To completely comprehend the toxic effects of the PLE, it may be necessary to compare several test models, including both normal and abnormal cells, for the purpose of determining cytotoxicity in the future.” was also added in the discussion in Section 3.6.

Reviewer 2 Report
Comments and Suggestions for Authors
In this manuscript, the authors characterised the leaves of three varieties of Thai papaya. They determined the basic chemical composition of the polyphenol content and determined the level of health-promoting activity through a number of determinants. The work is quite interesting from a scientific and application point of view, with the possibility of using raw materials for the production of nutraceuticals.
The introduction is comprehensive and introduces the topic of the work well; the aim is clear and transparent. The material and methods used are described in detail, but the following paragraph requires some supplementation:
line 113 what does proximate composition mean.
The structure of Table 2 requires supplementation.
The typical information on the characteristics of polyphenolic compounds is not provided. What spectra were used to identify individual compounds. The table should include data such as: m/z, ms/ms and Uv-Vis maxima. The compounds should be arranged in the order of elution. A chromatogram should also be attached.
I also suggest improving the manuscript in terms of statistical analysis. The number of results obtained defining various activities can be used to apply more modern statistical analyses, e.g. PCA or correlation. English needs correction
Comments on the Quality of English LanguageEnglish needs correction
Author Response
Reviewer 2
In this manuscript, the authors characterised the leaves of three varieties of Thai papaya. They determined the basic chemical composition of the polyphenol content and determined the level of health-promoting activity through a number of determinants. The work is quite interesting from a scientific and application point of view, with the possibility of using raw materials for the production of nutraceuticals.
The introduction is comprehensive and introduces the topic of the work well; the aim is clear and transparent. The material and methods used are described in detail, but the following paragraph requires some supplementation:
line 113 what does proximate composition mean.
Ans: The term "proximate composition," which is commonly used in the food and feed industry, refers to the six components of moisture, crude protein, fat, fiber, ash, and carbohydrates, each of which is given as a percentage of the sample.
The text originally indicated which components were evaluated in proximate composition. “The proximate composition of PL powder was determined using the AOAC [19] standard methods. Moisture (AOAC method number 950.46), crude protein (AOAC method number 928.08), ash (AOAC method number 920.153), fiber (AOAC method number 962.09), fat (AOAC method number 963.15), and carbohydrate (calculated by difference) were all assessed.”
The structure of Table 2 requires supplementation.
Ans: Done.
The typical information on the characteristics of polyphenolic compounds is not provided. What spectra were used to identify individual compounds. The table should include data such as: m/z, ms/ms and Uv-Vis maxima. The compounds should be arranged in the order of elution. A chromatogram should also be attached.
I also suggest improving the manuscript in terms of statistical analysis. The number of results obtained defining various activities can be used to apply more modern statistical analyses, e.g. PCA or correlation. English needs correction.
Ans: We have revised Table 2 as suggested. In addition, we have added chromatogram and applied PCA in Fig. 2.
Prof.Dr. Kalidas Shetty of North Dakota State University in the United States originally polished English. However, in the amended version, a paraphrase tool called QuillBot was employed to double-check the English.
A Pearson correlation analysis between phytochemical compositions and bioactivities was also performed.

Reviewer 3 Report
Comments and Suggestions for Authors
Ms. Ref. No.: Foods 3018202
Title: Phenolic and metabolic profiles, antioxidant activities, glycemic control, and anti-inflammatory activity of three Thai papaya cultivar leaves
The submitted manuscript is quite complete and includes an exhaustive analysis of the potential bioactivity of papaya leaf extracts.
The introduction is complete, the methodology is well described and the presentation of the results has been done in an orderly manner.
Author Response
Thank you very much.

Round 2
Reviewer 1 Report
Comments and Suggestions for Authors
I am happy with the revised version of the manuscript.
Author Response
Reviewer 1
I am happy with the revised version of the manuscript.
Ans: Thank you very much.
Reviewer 2 Report
Comments and Suggestions for Authors
In table 2 the value of m/z in negative mode for sinapic acid should be 223.
Author Response
Reviewer 2
In table 2 the value of m/z in negative mode for sinapic acid should be 223.
Ans: The value from the database match was 225.07565.